# Human Coronary Artery Endothelial Cell Response to *Porphyromonas gingivalis* W83 in a Collagen Three-Dimensional Culture Model

**DOI:** 10.3390/microorganisms12020248

**Published:** 2024-01-24

**Authors:** Andrés Cardona-Mendoza, Nelly Stella Roa Molina, Diana Marcela Castillo, Gloria Inés Lafaurie, Diego Fernando Gualtero Escobar

**Affiliations:** 1Grupo de Inmunología Celular y Molecular Universidad El Bosque-INMUBO, Vicerrectoría de Investigaciones, Facultad de Odontología, Universidad El Bosque, Bogota 11001, Colombia; acardonam@unbosque.edu.co; 2Unidad de Investigación Básica Oral-UIBO, Vicerrectoría de Investigaciones, Facultad de Odontología, Universidad El Bosque, Bogota 11001, Colombia; castillodiana@unbosque.edu.co (D.M.C.); lafauriegloria@unbosque.edu.co (G.I.L.); 3Centro de Investigaciones Odontológicas (CIO), Facultad de Odontología, Pontificia Universidad Javeriana, Bogota 110231, Colombia; nelly.roa@javeriana.edu.co

**Keywords:** endothelium, inflammation, endothelial dysfunction, *Porphyromonas gingivalis*, three-dimensional (3D) cell culture

## Abstract

*P. gingivalis* has been reported to be an endothelial cell inflammatory response inducer that can lead to endothelial dysfunction processes related to atherosclerosis; however, these studies have been carried out in vitro in cell culture models on two-dimensional (2D) plastic surfaces that do not simulate the natural environment where pathology develops. This work aimed to evaluate the pro-inflammatory response of human coronary artery endothelial cells (HCAECs) to *P. gingivalis* in a 3D cell culture model compared with a 2D cell culture. HCAECs were cultured for 7 days on type I collagen matrices in both cultures and were stimulated at an MOI of 1 or 100 with live *P. gingivalis* W83 for 24 h. The expression of the genes COX-2, eNOS, and vWF and the levels of the pro-inflammatory cytokines thromboxane A2 (TXA-2) and prostaglandin I2 (PGI2) were evaluated. *P. gingivalis* W83 in the 2D cell culture increased IL-8 levels at MOI 100 and decreased MCP-1 levels at both MOI 100 and MOI 1. In contrast, the 3D cell culture induced an increased gene expression of COX-2 at both MOIs and reduced MCP-1 levels at MOI 100, whereas the gene expression of eNOS, vWF, and IL-8 and the levels of TXA2 and PGI2 showed no significant changes. These data suggest that in the collagen 3D culture model, *P. gingivalis* W83 induces a weak endothelial inflammatory response.

## 1. Introduction

The vascular endothelium (VE) is versatile. It has diverse functions, such as the regulation of thrombosis, thrombolysis, and platelet adhesion, as well as the modulation of vascular tone, blood flow, and inflammatory immune responses [1]. The modulation or maintenance of vascular tone is primarily mediated by nitric oxide (NO), which activates guanylate cyclase in smooth muscle cells, thereby inducing vasodilation via cyclic guanosine monophosphate (cGMP) [2]. Prostaglandin I2 (PGI2), which is primarily derived from the action of cyclooxygenase-2 (COX-2) in arachidonic acid metabolism, is also known to modulate vascular tone via both cyclic adenosine monophosphate (cAMP)-dependent and cyclic adenosine monophosphate (cAMP)-independent pathways in smooth muscle cells [3]; in addition, together with NO, it helps to maintain an antithrombotic state of the VE [2], being the primary prostanoid inhibitor of platelet aggregation, thus becoming an essential atheroprotective factor [4] as it also inhibits leukocyte adhesion to the EV and acts as an anti-inflammatory mediator in the cardiovascular context [5].

In response to the different stimuli to which endothelial cells (ECs) are subjected, the transcription of genes associated with endothelial function, such as COX-2, endothelial nitric oxide synthase (eNOS), and von Willebrand factor (vWF), is activated or inhibited. These genes are involved in, among other functions, the pro-inflammatory or anti-inflammatory endothelial response. Therefore, maintaining endothelial function and genes associated with endothelial function is critical for preventing the development of cardiovascular diseases such as atherosclerosis (AT) [2]. The action of eNOS has been proposed as an atheroprotective factor, and the alteration of its expression has been shown to accelerate atherothrombotic processes in murine models [6,7]. COX-2 is involved in these cellular inflammatory processes by participating in the enzymatic biosynthesis of thromboxane A2 and PGI2, PGE2, PGD2, and PGF2a [8]. An increase in thromboxane A2 (TXA2) [9] is also observed in human atheromatous plaques and is associated with AT thrombotic problems [10], such as the induction of platelet aggregation [8]. Similarly, vWF is increased in individuals with AT, suggesting it is another possible factor associated with disease progression [11]; additionally, its release from the Weibel–Palade bodies results in platelet adhesion and aggregation following the binding with platelet glycoprotein [12]. Taken together, these alterations (among many others) help promote a state of endothelial dysfunction (ED) which, as initially stated, is the main characteristic of AT; it is characterized by a complex pathophysiology that is based on eNOS uncoupling and endothelial activation following abnormal gene expression from various inflammatory mediators as COX-2 is increased, vWF is released, and the downstream signaling via nuclear factor-kB (NF-kB) leads to the overexpression cytokines and adhesion molecules [12].

*Porphyromonas gingivalis* (*P. gingivalis*) is a Gram-negative anaerobic bacterium considered to be a keystone pathogen in periodontal disease, and in recent years, it has been associated with several systemic pathologies, including in [13,14,15]. This pathogen can pass into the bloodstream after daily oral hygiene activities, such as toothbrushing or periodontal treatments [16,17], which generate transitory bacteremia, allowing its transit through different mechanisms to the aortic endothelium [18], where it has been viably isolated from human atheromatous plaques [19,20]. Bacteremia with *P. gingivalis* is a factor that promotes inflammation and is therefore associated with ED, which is characterized by pro-inflammatory and prothrombotic states and processes that lead to atheromatous plaque formation [15]. Factors of virulence from *P. gingivalis*, such as gingipains, outer membrane vesicles (OMVs), and lipopolysaccharide (LPS), have shown the capacity to induce ED. LPS-Pg and bacterium induce the expression of angiotensin II and its receptor in HCAEC culture after repeated exposure [21]. Intercellular adhesion molecules such as PECAM-1 and VE-cadherin keep the endothelial permeability, and the integrity was affected when endothelial cells were exposed to *P. gingivalis* through a gingipain-dependent mechanism [22]. Endothelial microvesicles (EMVs) secreted by endothelial cells stimulated with *P. gingivalis* trigger the inflammatory endothelial response and altered eNOS and iNOS RNAm expression [23].

The role of *P. gingivalis* in endothelial activation/dysfunction has been studied in vitro for many years [24,25,26,27]. However, the applied models have some disadvantages because they have used human umbilical vein endothelial cells (HUVECs), which do not represent arterial endothelium physiology as they do not belong to the aortic vascular endothelium (VE); HUVECs have shown differences in their activation mechanisms and lipopolysaccharide (LPS)-induced responses compared with human coronary artery endothelial cells (HCAECs), which come from the aortic VE [28,29]. Atherosclerosis is induced in coronary artery endothelial cells when different factors alter the endothelial function within the wall of a large artery, causing a loss of intima integration [30]. The endothelial vascular system naturally grows on collagen proteins and glycoproteins constituting the intima. Type I collagens function like scaffolds and reservoirs of grown factors and different signaling powers for the endothelium, the clues necessary to eject the multiple functions of vascular contraction and dilation, permeability, catabolic metabolism, cellular traffic, and inflammatory response [31]. In this context, using HCAEC and a type I collagen scaffold as an in vitro cellular model represents one closer microenvironment for studying the pathogenesis of atherosclerotic cardiovascular diseases induced by periodontopathogens.

In addition, another disadvantage presented by these models is that they have been developed in conventional polystyrene tissue culture plates [24,25,26,27] (known as two-dimensional (2D) cell cultures). A 2D cell culture does not simulate the natural conditions of the VE, which is surrounded by an extracellular matrix (ECM) that it establishes a complex interaction with and that confers three-dimensional (3D) structural support that conditions its behavior and the consequent responses against any factor that damages the VE [32,33]. To provide more natural conditions to cell culture models and to accurately represent the tissue microenvironment, since the 1970s, 3D cell cultures that include several ECM components, such as collagen, have been developed, which allow for cell–ECM interactions and natural growth conditions and provide natural 3D structural support to the cells [34,35]. In this context, it has been shown that the behavior of the cells between 2D cell cultures vs. 3D cell cultures is different [26]. The behavioral changes of ECs depend on the type of ECM with which they interact [36] and their inflammatory responses to bacteria [37], showing that EC responses are conditioned by the cellular environment and highlighting the need to study them in vitro under more realistic conditions.

In a previous study by Gualtero et al. [38], we developed and characterized a culture system by seeding HCAECs onto 3D type I collagen, forming a functional endothelial monolayer. In this 3D model, the response induced by LPS-*P. gingivalis* (LPS-Pg) and LPS-*Aggregatibacter actinomycetemcomitans* (LPS-Aa) were evaluated and compared with the response of HCAECs in a conventional 2D polystyrene plate culture without collagen. This study showed that HCAECs in a 3D culture had a higher secretion profile of soluble factors (cytokines) than HCAECs in a 2D culture. In addition, a greater induction of the inflammatory response was observed in the 3D model than in the 2D model exposed to LPS-Aa. In contrast, an inflammatory response was not observed in the 2D and 3D models exposed to LPS-Pg. These results suggest differences in the mechanisms of pathogenesis induced by periodontopathogenic microorganisms, showing that the 3D model is a better model to evaluate the potential of *P. gingivalis* in causing ED. 

To date, the role of live *P. gingivalis* in the induction of ED in a 3D cell culture model with a biological ECM and HCAECs, two characteristics of the real context in which AT develops, has yet to be studied. Therefore, this work aimed to evaluate the response of HCAECs to *P. gingivalis* W83 in a 3D collagen cell culture model compared with the traditional 2D cell culture.

## 2. Materials and Methods

### 2.1. Preparation of Type I Collagen Matrices

Collagen matrices were prepared with bovine type I collagen at 5 mg/mL [39]. Type I collagen was proven using dispersion and isolated from the bovine fascia by the Grupo de Ingenieria de Tejidos from the Universidad Nacional de Colombia; an amount of 0.2 mg was deposited in 24-well plates and immediately frozen at −80 °C for 24 h to generate uniformity in the fiber distribution. Then, the matrices were lyophilized for 16 h. All plates were sterilized by ultraviolet (UV) irradiation for 15 min before use (Figure 1). Scaffolds elaborated with type I collagen have lamellar superficies and interconnected pores that allow the adhesion, proliferation, and apical growth of HCAEC, forming an endothelial monolayer on collagen matrices that mimic the intima in the vascular tissue. The 3D scaffold characterization was previously reported by Gualtero et al. [38].

### 2.2. Bacterial Culture

For all experiments, live *P. gingivalis* W83, considered one of the most pathogenic oral strains and highly virulent on HCAECs, was used [40]. The bacterial culture was carried out as described previously by our laboratory [41]. In brief, 30 µL of the cryopreserved strain was seeded on Brucella agar (BBL Microbiology Systems, Cockeysville, MD, USA) enriched with 0.3% *w*/*v* Bacto agar, 0.2% *w*/*v* yeast extract, 5% defibrinated sheep blood, 0.2% laked blood, 0.0005% hemin, and 0.00005% menadione and incubated in an anaerobic atmosphere with 9–13% CO_2_ and oxygen concentrations below 1% (Anaerogen, Oxoid, Hampshire, UK) at 36 °C for 7 days. The purity of the culture was also confirmed through microscopic characteristics with a negative UV light test and a positive CAAM test (CBZ-GLY-GLY-ARG 7-amido-4-methyl coumarin hydrochloride for the detection of trypsin-like enzymes) [42].

### 2.3. P. gingivalis Inoculum and Bacterial Viability

The inoculum was carried out with *P. gingivalis* W83 cultures on day 4 of incubation. As previously reported by Viafara-García et al. [21] in EGM-2MV medium (Lonza Walkersville Inc., Walkersville, MD, USA) enriched with growth factors without antibiotics, the inoculum was quantified by spectrophotometry (Thermo Scientific, Waltham, MA, USA) at the specific optical densities (ODs) of 0.900–0.908 at a wavelength of 620 nm; the purity conditions of the cultures were verified when the inoculum was performed and during the experiments. From this inoculum, multiplicities of infection (MOIs) of 1 (MOI 1) and 100 (MOI 100) were calculated.

The bacterial viability of the inoculum and bacteria in the supernatants 24 h after stimulation in the 2D and 3D cultures was confirmed with a viability mixture (Live/Dead^®^ BacLight Bacterial Viability, Thermo Fisher, Eugene, OR, USA) described by Castillo et al. [43]. In this experiment, it is essential to include a cell death control to corroborate the stability of the assay, which is why the bacterial inoculum treated with 0.2% chlorhexidine was included. The images were observed with a fluorescence microscope (Axio-Imager. A2. Zeizz^®^, Carl Zeiss Microscopy, LLC, White Plains, NY, USA) at a high magnification of 100× and digitized using AxioVision LE 4.8 software (Zeiss Microscopy); green fluorescence was used to observe the viable bacteria, and red fluorescence was used to observe the dead bacteria.

### 2.4. Stimulation of HCAECs with Live P. gingivalis W83

Primary cultures of HCAECs (Lonza Walkersville Inc., Walkersville, MD, USA) were seeded in T75 flasks (Corning, Glendale, CA, USA) at a density of 5000 cells/cm^3^ and incubated at 37 °C, 5% CO_2_, and 80% humidity in EGM-2MV medium (Lonza Walkersville Inc., Walkersville, MD, USA) enriched with growth factors. The medium was changed every 2 days until a confluence of close to 80% of the surface of the plate was observed. HCAECs at a density of 2 × 105 per well were seeded in 24-well plates directly on the plastic surface (2D culture) for 24 h until the formation of a 90% confluent monolayer; alternatively, they were seeded directly on top of the collagen support and cultivated for 7 days (3D cultures) until reaching monolayer confluence [38]. HCAECs were stimulated at MOI 1 and MOI 100 with 1 mL of live *P. gingivalis* W83 inoculum for 24 h in both the 2D and 3D cell cultures. HCAECs were stimulated with 1 μg/mL *A. actinomycetemcomitans* ATCC 29522-LPS (LPS-Aa) for 24 h as a positive control of the HCAEC inflammatory response [44].

#### *Aggregatibacter* *actinomycetemcomitans*

ATCC 29522, serotype b, was cultured in a microaerophilic environment on BHI agar (37 °C, 5% CO_2_, 72 h), and LPS-Aa was isolated by the phenol–water method, purified, and characterized as described [41]. HCAECs without stimuli were used as the no-stimulus control. 

### 2.5. Evaluation of the Cellular Viability of HCAECs Stimulated with P. gingivalis W83 in Both the 2D and 3D Cell Cultures

As previously described, HCAEC cells were seeded in 2D and 3D cell cultures and stimulated with *P. gingivalis* W83 at MOI 1 and MOI 100. Cell viability was evaluated by the resazurin reduction assay [45]. Unstimulated cells in both types of cell cultures were used as 100% cell viability controls. After 24 h of bacterial stimulation, the medium was removed, and the cells were washed with PBS; subsequently, the cells were incubated with a resazurin solution (0.44 μM) for 4 h. After incubation, fluorescence was measured at 530–590 nm using a plate reader (TECAN, INFINITE^®^ 200 PRO, Tecan Trading AG, Switzerland). Cell viability was expressed as the percentage of living cells relative to the control (without stimulus). Three independent experiments were performed, each with eight replicates.

### 2.6. Evaluation of eNOS, COX-2, and vWF Gene Expression in HCAECs Stimulated with live P. gingivalis W83

RNA was extracted from the HCAECs in the different study conditions in the 2D cell cultures by the phenol–chloroform method using TRIzol^®^ (Thermo Fisher Scientific, Waltham, MA, USA). To detach the cells from the collagen matrices, HCAEC 3D cell cultures were supplemented with 1 mL of collagenase at 1 mg/mL (Gibco, Life Technologies Corporation, Grand Island, NY, USA) for 45 min at 37 °C, and the RNA was subsequently extracted by the phenol–chloroform method using TRIzol^®^ (Thermo Fisher Scientific, Waltham, MA, USA).

Quantification of the relative expression of the endothelial function-related genes eNOS, COX-2, and vWF was performed by qRT-PCR with a qScript XLT 1-Step Kit (Quanta Bio, Beverly, MA, Germany) using the following primers: eNOS [46] forward: 5′-CCAGCTAGCCAAAGTCACCAT-3′, reverse: 5′-GTCTCGGAGCCATACAGGATT-3′; COX-2 [47] forward: 5′-AGGGTTGCTGGTGGTAGGAA-3′, reverse: 5′-GGTCAATGGAAGCCTGTGATACT-3′; vWF [48] forward: 5′-CACCATTCAGCTAAGAGGAGG-3′, reverse: 5′-GCCCTGGCAGTAGTGGATA-3; and GAPDH: forward: 5′-GGTGGTCTCCTCTGACTTCAACA-3′, reverse: 5′-GTTGCTGTAGCCAAATTCGTTGT-3′. The amplification protocol was as follows: one cycle at 95 °C for 5 min, followed by 35 cycles at 95 °C for 3 s and 60 °C for 30 cycles; then, after 40 °C for 1 min, a melting curve from 65 °C to 95 °C at increments of 0.5 °C was performed. In all of the analyses, 10 ng/µL RNA was used. The expression of each gene was normalized to that of the housekeeping gene GAPDH within each treatment. The relative expression of the gene of interest in each experimental condition was calculated relative to the control without stimulus using the ΔΔCq method with Bio–Rad CFX manager 3.1 software (Bio–Rad, Hercules, CA, USA). Each experimental condition was analyzed in duplicate for three independent experiments.

### 2.7. Quantification of Pro-Inflammatory Cytokines and Chemokines Produced by HCAECs Stimulated with Live P. gingivalis W83

The supernatants of both the 2D and 3D cell cultures in the different experimental conditions were collected. The levels of the pro-inflammatory cytokines interleukin (IL)-1α, IL-1β, IL-6, and tumor necrosis factor (TNF)-α and the chemokines RANTES, MCP-1α, MIP-1α, and IL-8 were measured by flow cytometry with a LEGENDplex™ kit (BioLegend, San Diego, CA, USA). The data acquisition was performed with a BD FACS Accuri™ C6 Plus, and the data processing was performed using LEGENDplex software v8.0 (Biolegend, San Diego, CA, USA). Each experimental condition was analyzed in duplicate for three independent experiments.

### 2.8. Quantification of the Levels of Thromboxane A2 (TXA2) and Prostaglandin I2 (PGI2) Produced by HCAECs Stimulated with Live P. gingivalis W83

The supernatants of both the 2D and 3D cell cultures in the different experimental conditions were collected, and the levels of TXA2 and PGI2 were measured by competitive ELISA using a thromboxane B2 kit (Cayman Chemical, Ann Arbor, MI, USA) and a 6-keto prostaglandin F1α kit (Cayman Chemical, Ann Arbor, MI, USA), respectively. The data were acquired with an Infinite 200 PRO reader (TECAN). For the data analysis, the manufacturer’s recommendations were followed. Each experimental condition was analyzed in duplicate for three independent experiments.

### 2.9. Data Analysis

Shapiro–Wilks tests were used for the comparative analysis among the study conditions regarding the expression of genes, cytokines, and prostanoids. Kruskal–Wallis tests were used to compare all conditions, and Mann–Whitney U tests were used for pairs. Differences were considered statistically significant when *p* < 0.05. 

## 3. Results

### 3.1. Effect of live P. gingivalis W83 on the Viability of HCAECs

First, we evaluated bacterial viability before and after stimulation. As shown in Figure 2A, the viability of the bacteria was confirmed at the time of HCAEC stimulation, and 100% bacteria viability was observed after 24 h of HCAEC stimulation in both the MOI 1 and MOI 100 conditions (Figure 2C–F).

Additionally, we evaluated the effect of the live bacteria on cell viability and validated the MOIs. HCAECs were stimulated with live *P. gingivalis* W83 at MOI 1 and MOI 100 for 24 h in both 2D and 3D cell cultures, and cell viability was evaluated using the resazurin assay. As observed in Figure 2G, the cell viability percentage of the cells stimulated at MOI 1 in the 2D cell cultures was 87.93%, and that in the 3D cell cultures was 81.76%. On the other hand, the cell viability percentage of the cells stimulated at MOI 100 in the 2D cell cultures was 96.22%, and that in the 3D cell cultures was 91.60%. The reduction in the percentage of cell viability in all cases was not significant (*p* > 0.05). Therefore, we can conclude that stimulation with live *P. gingivalis* W83 at MOI 1 and MOI 100 for 24 h does not affect cell viability. Both are optimal MOIs for the experimental conditions. Together, we can conclude that at both MOI 1 and MOI 100, the viability of HCAECs is maintained when the bacterial stimulus is live bacteria.

### 3.2. P. gingivalis W83 Increased COX-2 Gene Expression in HCAECs in the 3D Cell Culture Model

Endothelial function-associated gene transcription is activated or inhibited in response to several stimuli to which ECs are subjected. To evaluate whether *P. gingivalis* alters COX-2, eNOS, and vWF gene expression in vitro in 3D cell cultures, HCAECs on bovine type I collagen matrices were stimulated with live *P. gingivalis* W83 at MOIs of 100 and 1, taking the 2D traditional cell culture as the comparative model (Figure 3). Figure 3a,b shows the relative expression of COX-2 after 24 h of HCAEC infection with *P. gingivalis* W83 in both the 2D and 3D cell cultures. In the 2D cell cultures, as shown in Figure 3a, no significant differences were observed in the expression of COX-2 at MOI 100 and MOI 1 compared with the control without stimulus (Control) (*p* = 0.1667), which was only stimulated with LPS-Aa. In contrast, in 3D cell cultures stimulated at MOI 100 (*p* = 0.0139) and MOI 1 (*p* = 0.0139), the expression of COX-2 was significantly higher compared with the control, with stimulation at MOI 1 showing the highest increase (Figure 3b). On the other hand, stimulation with *P. gingivalis* W83 did not significantly change the eNOS or vWF gene expression in HCAECs in both the 2D and 3D cell cultures compared with the control (*p* > 0.05), as shown in Figure 3c–f. It should be noted that in the 2D cell culture, the stimulation of HCAECs at MOI 100 increased eNOS expression; however, this increase did not reach statistical significance (Figure 3c).

### 3.3. P. gingivalis W83 Reduced the Levels of MCP-1 in HCAECs in the 2D and 3D Cell Cultures, While the Levels of IL-8 Were Maintained in the 3D Cell Cultures

In the endothelial response, pro-inflammatory cytokine and chemokine production are critical to promoting leukocyte activation and adhesion to ECs. To evaluate the in vitro effect of *P. gingivalis* on pro-inflammatory cytokine and chemokine production in the HCAEC response to this bacterium, the supernatants of the 3D cell cultures in the different conditions were collected, and IL-1α, IL-1β, IL-6, TNF-α, RANTES, MCP-1α, MIP-1α, and IL-8 levels were measured by flow cytometry. Only MCP-1, IL-6, and IL-8 were detected in the whole panel. The IL-8 levels in the 2D cultures stimulated at MOI 100 with *P. gingivalis* W83 showed a significant increase (*p* = 0.0039) compared with the control (Figure 4a). Unlike the 2D cultures, IL-8 levels did not show significant variations at MOI 100 (*p* = 0.7488) or MOI 1 (*p* = 0.2002) compared with the control (Figure 4b). Interestingly, in the 2D cell cultures, it was observed that the levels of MCP-1 produced in HCAECs decreased at MOI 100 (*p* = 0.0374) and MOI 1 (*p* = 0.0065) when stimulated with *P. gingivalis* W83 (Figure 4c). Similarly, MCP-1 levels decreased in the 3D cell cultures stimulated at MOI 100 (*p* = 0.0065), as shown in Figure 4d. Likewise, IL-6 levels decreased considerably when stimulated at MOI 100 compared with the control levels, although the difference was not statistically significant (Figure 4f). As expected, LPS-Aa significantly increased MCP-1, IL-6, and IL-8 levels compared with the control in all conditions evaluated. The soluble levels of IL-8, MCP-1, and IL-6 in the controls were higher in the 3D cell cultures than in the 2D cell cultures.

### 3.4. P. gingivalis W83 Tended to Increase the Levels of PGI2 and TXA2 in HCAECs in 3D Cell Cultures

Because COX-2 is an enzyme whose action is involved in the biosynthesis of prostanoids such as TXA2 and PGI2, which mediate inflammatory processes, we evaluated whether *P. gingivalis* W83 induced changes in the PGI2 and TXA2 levels in vitro in HCAECs. The supernatants of both cell cultures in different conditions were collected and analyzed by competitive ELISA to determine the levels of these prostanoids. Figure 5 shows that the variations in the levels of both TXA2 and PI2 at MOI 100 and MOI 1 were not significantly different from the control in either the 2D or 3D cell cultures. However, in the 2D cell cultures, there was a trend toward an increase compared with the control PGI2 levels (Figure 5c). Similarly, a trend was observed in the 3D cell cultures with TXA2 and PGI2 at MOI 100 and MOI 1 (Figure 5b,d).

## 4. Discussion

In this study, human coronary artery endothelial cells were cultured in 2D and 3D models to determine the response related to the ED front to *P. gingivalis W83*. The results showed that *P. gingivalis* W83 poorly induces an inflammatory response in both models. However, the 3D model showed altered secretion of thromboxane A2 and prostaglandin I2 compared with the 2D model. Therefore, the HCAEC culture on the type I collagen scaffold provides one platform that gives information not detectable in 2D models to elucidate the role of periodontopathic bacterium in ED and atherosclerosis.

Although the possibility of the action of gingipains in the degradation of cytokines cannot be ruled on, these results are consistent with previous studies where HCAECs showed a poor inflammatory response to live *P. gingivalis* W83 with low IL-8 and MCP-1 expression when the cells were exposed to repeated doses of *P. gingivalis* [21]. Similar results were observed in 2D and 3D endothelial cultures and stimulated with LPS-Pg [28]. *P. gingivalis* can evade the immune system and has been isolated from human atherosclerotic plaques [49]. *P. gingivalis* W83 is characterized in vitro by its high capacity to adhere, invade, and persist in HAECs [40], properties that have been associated with the induction of ED [15]; therefore, it is an excellent candidate strain to study the mechanisms by which this periodontopathogen is associated with atherogenic processes. Our results support the hypothesis that *P. gingivalis* does not activate the endothelium, which may be a form of reducing the inflammation and immunologic cellular response and inducing a chronic and subclinical CVD disease. Although *P. gingivalis* has been detected and isolated from atherosclerotic lesions [19], it probably does not activate the inflammatory response of ECs as an evasion mechanism to infect the endothelium and generate ED through chronic infection.

Similar to that previously reported by Chou et al. [50] with *P. gingivalis* 381, in this work, *P. gingivalis* W83 induced an increase (not significant) in the gene expression of COX-2 in HCAECs in the 2D cell culture model at MOI 100 (Figure 3a), despite the differences in the strains used between these two studies. Surprisingly, the greatest increase in COX-2 expression was in 3D cell cultures at MOI 1 (Figure 3b), suggesting a role for collagen in the expression of COX-2 in HCAECs in response to *P. gingivalis* W83. This subject will be the object of study in future research.

Different studies [51,52] have confirmed that the type of fimbriae of the various strains of *P. gingivalis* is widely related to the immunoregulatory capacity, which is why even though strains 381 and 33,277 are 99% genetically homologous, there is a change in the fimB allele of the 33,277 strain in addition to finding greater gingipain activity on the surface of the cells of this strain, which may make it less pro-inflammatory. The W83 strain, due to its type of short fimbriae (Fim IV) and the presence of a capsule, stimulates the inflammatory response to a lesser extent.

eNOS is an atheroprotective factor [6] with anti-inflammatory actions due to nitric oxide synthesis. The gene expression of eNOS in the HCAECs in the 2D cell cultures was increased (without being significant) at MOI 100 (Figure 3c). In other models, infection of HUVECs with *P. gingivalis* W83 [53] and ATCC 33277 [54] reduced the soluble levels of eNOS mediated by the induction of the activation of PPARγ in ECs [53] and the action of bacterial HSP 60 [55]; however, in the model in this work, these data cannot be adequately related as the soluble concentration of eNOS was not measured. This discordance of eNOS expression between HUVEC and HCAEC stimulated with *P. gingivalis* shows the relevance of using the pertinent cellular type where atherosclerosis is developed. One previous study showed that when endothelial cells were isolated from the human umbilical veins, arteries, and coronary arteries and were exposed to diverse stimuli, they altered the expression of adhesion molecules, eNOS, and cytokines and the migration of neutrophils [56]. In this context, the sources of the cellular phenotype contribute to differences in the responses of ECs in different locations in the vasculature.

As mentioned above, the MCP-1 levels were reduced by *P. gingivalis* in both types of cell cultures, an effect that could have been due to the proteolytic action of gingipains, as previously reported with *P. gingivalis* 381 in HUVECs [57]; however, this result is contrary to what was reported by Rodriguez et al. [40] with *P. gingivalis* W83 at MOI 100 in human aortic endothelial cells (HAECs). This discrepancy is probably due to the type of ECs used in the different studies. Similar to our results, in a previously reported model with *P. gingivalis* W83 LPS [38], the MCP-1, IL-6, and IL-8 levels showed no significant changes compared with those in the control.

Contrary to expectations, *P. gingivalis* W83 has been described to induce a weak inflammatory response in ECs compared with the response observed with other ATCC strains, such as 33,277 and 381 [40], a capacity attributed to the presence of a capsule with the K1 capsular antigen in W83, which reduces the inflammatory response induction in other cells, such as human gingival fibroblasts [58] and macrophages [59]. Accordingly, Brunner et al. [58] showed that W83’s capsule inhibits IL-8 gene expression in human gingival fibroblasts. This could explain why the IL-8 levels produced by HCAECs in this work did not vary, which was a striking result.

IL-6 plays an essential role in fatty streak formation, increasing the proliferation and migration of smooth muscle cells and the induction of acute-phase protein synthesis, events related to the atherogenic process [60]. Interestingly, the IL-6 levels diminished strongly at MOI 100, although the changes were not statistically significant. Similar to IL-8, Brunner et al. [58] also showed that W83’s capsule inhibited IL-6 gene expression, which could explain our results.

This result can be understood by the presence of the collagen ECM that provided HCAECs with a tissue microenvironment that conditioned them to be less responsive, as would be expected in the natural environment. Type I collagen is a protein that is present as an ECM component and plays different cellular roles, mainly, the support of tissue and the molecular diffusion of sign and growth factors. The vascular tissue is conformed by three layers: the endothelium, intima, and media. The endothelium is constituted by one monolayer of endothelial cells, where they rest on glycoprotein, type IV and I collagens, and other proteins that conform to the intima. In contrast, the layer media have smooth muscular cells, fibroblasts, and collagen proteins like type I collagen [61]. Therefore, type I collagen is one abundant protein in vascular tissue, and the ECM protein is usually used in engineering tissue like scaffolds or gels and in vascular medicine like vascular grafts [62,63]. Other cellular models use Matrigel as substrates to induce endothelial proliferation and angiogenesis [38]. In some cases, cells in Matrigel are used like vascular models, but this does not represent the structure on which the endothelial cells grow and form a monolayer in the endothelium vascular. The type I collagen scaffold provides a structure that promotes endothelial cells’ growth and monolayer formation and modulates their inflammatory response to LPS and bacteria from periodontopathogens [38,64].

On the other hand, the capsule of *P. gingivalis* W83 as a virulence factor likely had an essential role in inhibiting the inflammatory endothelial response. In addition, EMVs secreted by ECs after *P. gingivalis* stimulation affect the cellular permeability mediated by gingipains [22]. To continue evaluating the endothelial response to *P. gingivalis* in models with an ECM, it is recommended that another strain without a capsule and with a high capacity to induce an inflammatory response, such as *P. gingivalis* 381 [33], be evaluated. We have developed a 3D endothelial model that reflects some aspects of the endothelium and intima vascular; upcoming research should include dynamic and fluidic stimulation to mimic the blood-induced shear stress. The previous conditions would elucidate how oral pathogens interact with endothelial cells and other immune cells to orchestrate different ways of ED that carry atherosclerosis. *P. gingivalis* and their virulence factors affect the endothelium with other bacteria, enhancing inflammatory response, which is another aspect to research regarding 3D endothelial fluidic models and multiple oral pathogens. In addition, this 3D endothelial model could be used to evaluate potential target therapeutic to reduce vascular injury caused by periodontopathogens.

Within the limitations of this study, the results support that *P. gingivalis* W83 is a poor inductor of inflammation on the endothelium, and there may be other ways of ED that would be active for inducing atherosclerotic cardiovascular disease. As insights for the future, additional investigations using 3D vascular models will reveal how other oral bacteria periodontopathically impact cardiovascular health or favor the endothelial pro-inflammatory response.

## Figures and Tables

**Figure 1 microorganisms-12-00248-f001:**
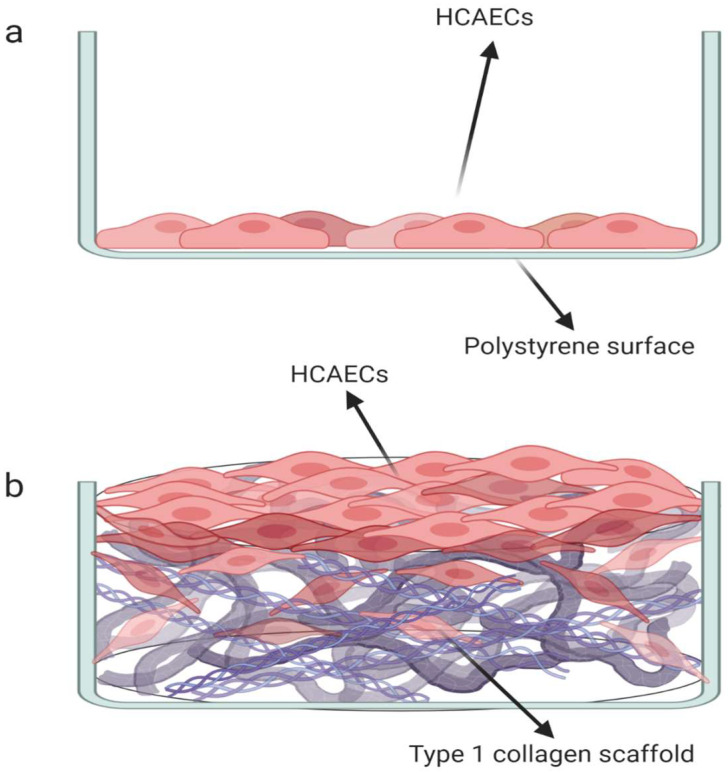
(**a**) Two-dimensional cell culture model. (**b**) Three-dimensional cell culture model.

**Figure 2 microorganisms-12-00248-f002:**
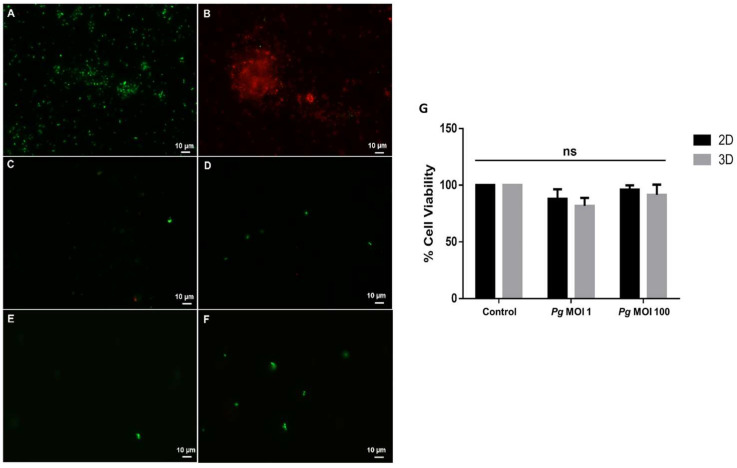
Cell viability of HCAECs stimulated with *P. gingivalis* W83 for 24 h at M OI 1 and MOI 100. (**A**–**F**) Green fluorescence indicates the live bacteria and red fluorescence indicates the dead bacteria. (**A**) Bacterial inoculum adjusted for stimulation of HCAECs; (**B**) *P. gingivalis* W83 was treated with 0.2% chlorhexidine as a control to create dead cells; (**C**) HCAECs in the 2D culture were stimulated with *P. gingivalis* W83 at MOI 1; (**D**) HCAECs in the 2D culture were stimulated with *P. gingivalis* W83 at MOI 100; (**E**) HCAECs in the 3D culture were stimulated with *P. gingivalis* at MOI 1; (**F**) HCAECs in the 3D culture were stimulated with *P. gingivalis* W83 at MOI 100; (**G**) The cell viability of HCAECs stimulated with *P. gingivalis* W83 at MOI 1 and MOI 100 in 2D and 3D cell cultures. Control: control without stimulus. MOI 100: a multiplicity of infection for *P. gingivalis*/HCAECs of 100:1. MOI 1: a multiplicity of infection for *P. gingivalis*/HCAECs of 1:1. *p* < 0.05 compared with the control, ns: Not significant. The data are represented by the median of each group of three independent experiments with eight replicates.

**Figure 3 microorganisms-12-00248-f003:**
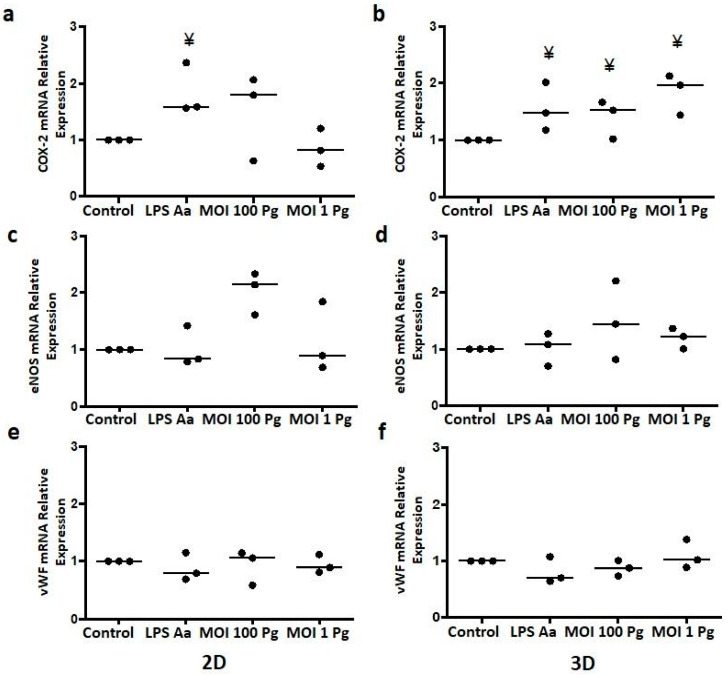
Relative COX-2, eNOS, and vWF genes expression in HCAECs at 24 h post stimulation with *P. gingivalis* W83 in 2D and 3D cell cultures as determined by qRT-PCR. (**a**,**b**) COX2 mRNA relative expression; (**c**,**d**) eNOS mRNA relative expression; (**e**,**f**) vWF mRNA relative expression. Control: control without stimulus, LPS-Aa: lipopolysaccharide from *Aggregatibacter actinomycetemcomitans* ATCC 29522 (1 µg/mL). MOI 100: a multiplicity of infection for *P. gingivalis*/HCAECs of 100:1. MOI 1: a multiplicity of infection for *P. gingivalis*/HCAECs of 1:1. ¥ *p* < 0.05 compared with the control. The data are represented by the median of each group of three independent experiments conducted in duplicate.

**Figure 4 microorganisms-12-00248-f004:**
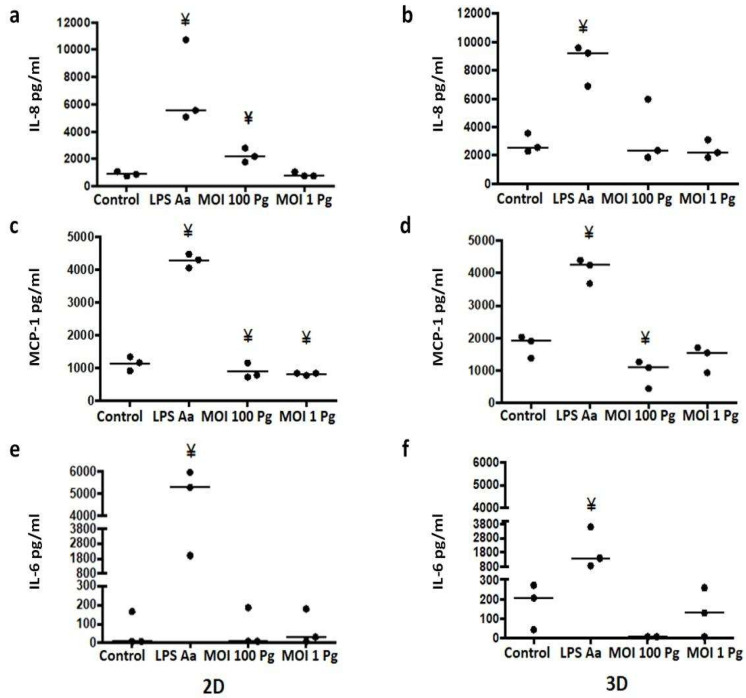
As measured by flow cytometry, the levels of IL-8, MCP-1, and IL-6 in HCAECs at 24 h post stimulation with *P. gingivalis* W83 in the 2D and 3D cell cultures. (**a**,**b**) IL-8 secretion; (**c**,**d**) MCP-1 secretion; (**e**,**f**) IL-6 secretion. Control: control without stimulus. LPS-Aa: lipopolysaccharide from *Aggregatibacter actinomycetemcomitans* ATCC 29522 [1 µg/mL]. MOI 100: a multiplicity of infection for *P. gingivalis*/HCAECs of 100:1. MOI 1: a multiplicity of infection for *P. gingivalis*/HCAECs of 1:1. ¥ *p* < 0.05 compared with the control. The data are represented by the median of each group of three independent experiments conducted in duplicate.

**Figure 5 microorganisms-12-00248-f005:**
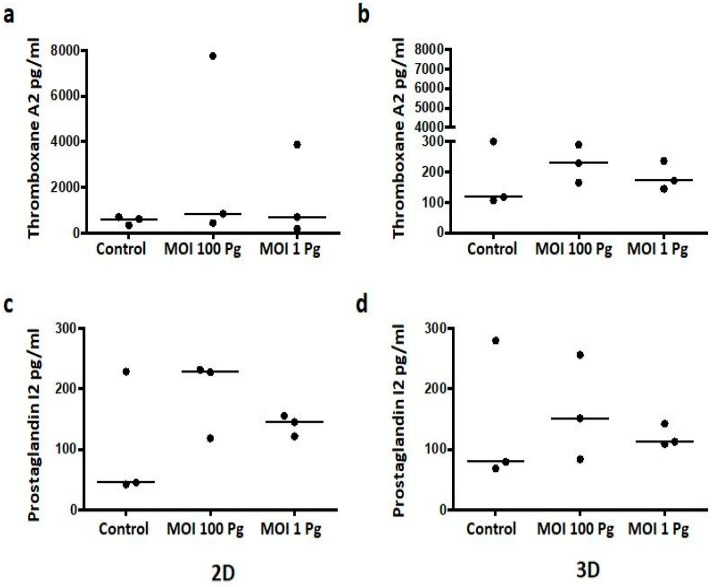
As measured by ELISA, the thromboxane A2 and prostaglandin I2 levels in HCAECs were measured at 24 h post stimulation with *P. gingivalis* W83 in the 2D and 3D cell cultures. (**a**,**b**) Thromboxane A2 secretion; (**c**,**d**) prostaglandin I2 secretion. Control: control without stimulus. MOI 100: a multiplicity of infection for *P. gingivalis*/HCAECs of 100:1. MOI 1: a multiplicity of infection for *P. gingivalis*/HCAECs of 1:1. The data are represented by the median of each group of three independent experiments conducted in duplicate.

## Data Availability

Data are contained within the article.

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
