# Peer review of "Human Coronary Artery Endothelial Cell Response to Porphyromonas gingivalis W83 in a Collagen Three-Dimensional Culture Model"

_microorganisms, 2024, doi:10.3390/microorganisms12020248_

Round 1

Reviewer 1 Report

Comments and Suggestions for Authors

see attached file

Comments on the Quality of English Language

very minor syntax/word choice improvements noted in comments.

Author Response

The following is a response to the reviewers' corrections and suggestions. We thank them for their time and comments; we are convinced that it will improve the article for publication. All corrections are in red in the text of the article.

 Response to reviewer 1

Line 49: Please capitalize "Gram."

Response:  Corrected in the text

Lines 51-55 and 339-340: To the reviewer's knowledge, recovery of viable P. gingivalis from atheromatous plaques has not been reliably reported. This is a very "live" issue in the field of periodontal microbiology. The two reports cited are suggestive, but inconclusive at best. Atarbashi-Moghadam et al. reported detection by PCR of P. gingivalis DNA in atherosclerotic plaque of 2/21 subjects with both coronary artery disease and periodontitis, but did not report detection of viable (or even intact) P. gingivalis. Kozarov et al detected P. gingivalis (by monoclonal antibody) associated with ECV-304 cells (a tumorigenic cell line of disputed provenance) that had been co-cultured with homogenized carotid atherosclerotic plaque tissue from a single subject.

  • In the case of the work of Atarbashi-Moghadam et al., it was a mistake on our part. But in the work of Kozarov et al. If the recovery of viable bacteria can be argued, since they performed invasion assays on non-phagocytic ECV-304 cells (regardless of the problems of origin), These invasion assays were positive, and for Pg and Aa to invade cells, they must be viable. However, the word "viable" is removed from line 56 to only refer to the isolation of bacteria in atheroma in both studies.
  • On lines 339-340 there is nothing about the recovery of viable bacteria from atheromas.
  • The word "viable" is removed from line 349

Line 84: Please italicize species names.

Response:   Corrected in the text

Line 136: The more common term is “laked” rather than “lacquered.”

Response:  Corrected in the text

Line 143: What does this sentence mean? “realized”?

Response:  Corrected in the text: "realized" is changed to "carried out."

Figure 2 a-f: This figure is very hard to see either in print or on-screen

Response:  The size of the photos was enlarged (Figure 2a-f)

Lines 81-92: Numerous prior studies have utilized Matrigel as a substrate for the growth of cell cultures for challenge/response studies. It would be appropriate to note here and in the Discussion why the present system using Type I collagen is preferable to a 3D model using Matrigel.

Response: Thank you for your suggestion; we have included this topic in the discussion

Methods, line 100: Please indicate the source of the collagen used in this study.

Response: we have indicated the sources of collagen in the methods section

Lines 166-167: How was A. actinomycetemcomitans grown? How was its LPS isolated? Please provide details or citations for this important control used in Figures 4 and 5.

 Response:  The culture and LPS isolated from A.a was cited in the methods.

Line 173: Please provide a source/reference for the resazurin reduction assay used to assay the viability of HCAEC.

 Response: the reference was incorporated into the text in methods ref :

Kumar, P., Nagarajan, A., & Uchil, P. D. (2018). Analysis of Cell Viability by the alamarBlue Assay. Cold Spring Harbor protocols, 2018(6), 10.1101/pdb.prot095489. https://doi.org/10.1101/pdb.prot095489

Lines 229-232: The Live/Dead assay is essentially a measure of cell membrane integrity. While membrane integrity is essential for viability, it is not a sufficient condition for fastidious microbes like P. gingivalis, which typically enter a “viable but non-culturable” state after 24 h incubation under ideal conditions (appropriate medium, anoxic

 atmosphere).

 Response:

Thank you very much for your comment, as you indicate, the Live/Dead assay bacterial viability evaluation method is based on the integrity of the bacteria membrane, but it is an assay that is validated to indicate with certainty that bacteria with green fluorescence are found viable, this type of assay has been previously used in different published research by our group and other researchers:

Some of the bibliographic references in which this test has been used to evaluate viability are:

  • Aherne O, Ortiz R, Fazli MM, Davies JR. Effects of stabilized hypochlorous acid on oral biofilm bacteria. BMC Oral Health. 2022 Sep 20;22(1):415. doi: 10.1186/s12903-022-02453-2. PMID: 36127658; PMCID: PMC9487106.

  • Lai Y, Xu Z, Chen J, Zhou R, Tian J, Cai Y. Biofunctionalization of Microgroove Surfaces with Antibacterial Nanocoatings. Biomed Res Int. 2020 Jun 17;2020:8387574. doi: 10.1155/2020/8387574. PMID: 32626766; PMCID: PMC7317309.

  • Martínez-Hernández M, Reyes-Grajeda JP, Hannig M, Almaguer-Flores A. Salivary pellicle modulates biofilm formation on titanium surfaces. Clin Oral Investig. 2023 Oct;27(10):6135-6145. doi: 10.1007/s00784-023-05230-9. Epub 2023 Aug 30. PMID: 37646908; PMCID: PMC10560156.

  • Gutiérrez DM, Castillo Y, Ibarra-Avila H, López M, Orozco JC, Lafaurie GI, Castillo DM. A new model for the formation of an Enterococcus faecalis endodontic biofilm with nutritional restriction. J Basic Microbiol. 2022 Jan;62(1):13-21. doi: 10.1002/jobm.202100533. Epub 2021 Dec 7. PMID: 34874074.

  • Lafaurie GI, Zaror C, Díaz-Báez D, Castillo DM, De Ávila J, Trujillo TG, Calderón-Mendoza J. Evaluation of substantivity of hypochlorous acid as an antiplaque agent: A randomized controlled trial. Int J Dent Hyg. 2018 Nov;16(4):527-534. doi: 10.1111/idh.12342. Epub 2018 Apr 2. PMID: 29608039.

  • Castillo DM, Castillo Y, Delgadillo NA, Neuta Y, Jola J, Calderón JL, Lafaurie GI. Viability and Effects on Bacterial Proteins by Oral Rinses with Hypochlorous Acid as Active Ingredient. Braz Dent J. 2015 Oct;26(5):519-24. doi: 10.1590/0103-6440201300388. PMID: 26647939.

Additionally, we have confirmed the viability after exposure time to different cell models in different works.

  1. gingivalis is an anaerobic bacteria, which, under the right conditions (as explained in the text), grows easily as has been previously demonstrated:

  • Pianeta R, Iniesta M, Castillo DM, Lafaurie GI, Sanz M, Herrera D. Characterization of the Subgingival Cultivable Microbiota in Patients with Different Stages of Periodontitis in Spain and Colombia. A Cross-Sectional Study. Microorganisms. 2021 Sep 12;9(9):1940. doi: 10.3390/microorganisms9091940. PMID: 34576835; PMCID: PMC8469102.
  • Mayorga-Fayad I, Lafaurie GI, Contreras A, Castillo DM, Barón A, Aya Mdel R. Microflora subgingival en periodontitis crónica y agresiva en Bogotá, Colombia: un acercamiento epidemiológico [Subgingival microbiota in chronic and aggressive periodontitis in Bogotá, Colombia: an epidemiological approach]. Biomedica. 2007 Mar;27(1):21-33. Spanish. Epub 2007 May 31. PMID: 17546221.

This bacteria is anaerobic, but it has been shown that it can resist cell culture conditions for up to 48 hours in a viable manner in different cellular experiments, as previously published:

  • Chen R, Ji Y, Li T, Zhao B, Guo H, Wang Z, Yao H, Zhang Z, Liu C, Du M. Anti-Porphyromonas gingivalis nano therapy for maintaining bacterial homeostasis in periodontitis. Int J Antimicrob Agents. 2023 Jun;61(6):106801. doi: 10.1016/j.ijantimicag.2023.106801. Epub 2023 Apr 3. PMID: 37019242.

  • Minne X, Mbuya Malaïka Mutombo J, Chandad F, Fanganiello RD, Houde VP. Porphyromonas gingivalis under palmitate-induced obesogenic microenvironment modulates the inflammatory transcriptional signature of macrophage-like cells. PLoS One. 2023 Jun 29;18(6):e0288009. doi: 10.1371/journal.pone.0288009. PMID: 37384642; PMCID: PMC10309636.

Like these works, there are many more that confirm this fact.

Figure 2 legend, Lines 248-249: Please describe chlorhexidine treatment control in Methods section. It would be appropriate to include the challenge of HCAEC with killed P. gingivalis (either chlorhexidine- or heat-treated)

Response:  Within the bacterial viability experiment, viability controls and mortality controls must be used. Based on the group's previous experience in laboratory conditions, 0.2% chlorhexidine is used as a cell death control, as can be seen in the image of the bacteria treated with this substance. They have 100% mortality. Se aclara en los metodos

Lines 367-373: While P. gingivalis strains differ in several important ways, the authors do not include any data derived from strains other than W83, nor any data indicating a contribution of gingipain to any of the observed effects on HCAEC. This reviewer suggests minimizing this part of the Discussion.

The discussion on this subject is minimal and only similarities with other strains of the bacterium are raised. As for gingipains, no effect is attributed to them, it is only said that "their action cannot be ruled out" as an important virulence factor, considering that we worked with live bacteria.

Line 147: How was the relationship between optical density and cell number determined

Response:

As previously indicated, the CFU ratio was previously standardized (Viafara-Garcia et al. 2019).

For this, includes of P. gingivalis were carried out in a BHI broth of bacteria with 4 days of incubation under anaerobiosis conditions. The optical densities were measured in a spectrophotometer, and they were plated on modified brucella agar as previously clarified. Eight days later, the CFU counts were carried out to confirm the number of bacteria for each experiment was previously standardized under laboratory conditions to obtain 2.6 X 109 bacteria/mL.

Reviewer 2 Report

Comments and Suggestions for Authors

 The main objective of this study was to investigate the impact of the interaction between the periodontal pathogen Porphyromonas gingivalis and human coronary artery endothelial cells (HCAECs) in different cell culture models. The study focused on evaluating the effects of the bacterium in terms of cell viability, gene expression of factors associated with endothelial function, and the production of pro-inflammatory cytokines and chemokines. The aim was to better understand the inflammatory and immunomodulatory responses of endothelial cells in response to the presence of this pathogenic bacterium, providing crucial insights into the implications of P. gingivalis infection in the context of cardiovascular diseases.

Below, I provide several suggestions aimed at enhancing the quality of the research.

Introduction

The introduction provides a concise overview of genes associated with endothelial function and their role in atherosclerosis (AS), while highlighting the potential role of Porphyromonas gingivalis (P. gingivalis) in endothelial dysfunction (ED). However, to improve the introduction, here are some suggestions:

1. Elaborate further on how exactly COX-2, eNOS, and vWF genes influence endothelial function and how their alterations can lead to ED. This could help strengthen the link between genes and atherosclerosis.

2. Expand the discussion on the relationship between P. gingivalis and endothelial dysfunction. Explain in more detail how the presence of P. gingivalis is directly associated with endothelial dysfunction, highlighting the exact mechanisms by which this bacterium promotes inflammation and atherosclerotic plaque formation.

3. Explicitly reinforce the limitations of previous cell culture models using HUVECs, emphasizing the importance of using HCAECs in a more representative model of the atherosclerosis context.

4. Emphasize the benefits of 3D models. Explain in more detail how the 3D model with collagen allows for a more accurate representation of the microenvironment and cell-matrix interactions, highlighting why it is crucial to study endothelial responses in a more realistic environment.

5. Clearly highlight the specific objective of this study, which is to evaluate the response of HCAECs to P. gingivalis W83 in a 3D cell culture model with collagen compared to the traditional 2D culture model.

Materials and Methods

1. The preparation procedures seem detailed and well-defined, allowing for experiment replication. However, detailed characterization of collagen matrices was not provided in this excerpt, despite referencing a previous study for this purpose. It would be advisable to include more details about this characterization in this section.

2. The bacterial culture conditions are well-described, indicating the medium, concentrations, and incubation conditions. However, quantitative data on culture purity are not provided. How did the authors assess this? It would be helpful to include information on contamination, if any, and the percentage of purity.

3. The quantification of the P. gingivalis inoculum is described, indicating quantification methods by optical density. However, presenting numerical results of these optical densities would allow for a clearer assessment of bacterial concentration. The evaluation of bacterial viability after 24 hours of stimulation in 2D and 3D cultures is a good indicator, but quantitative results of this viability are not mentioned.

4. The methods for culturing human coronary artery endothelial cells (HCAECs) are well-detailed, specifying culture conditions and cell density. However, it is not mentioned whether the experiments were blinded or randomized to minimize bias.

Results

1. Remove unnecessary repetitions to simplify data presentation.

2. When mentioning statistically significant or nonsignificant results (p < 0.05 or p > 0.05), it is essential to include the name of the statistical test used to provide a solid basis for conclusions.

Discussion

1. The discussion should be organized into clear sections addressing different aspects of the results, such as inflammatory response, gene expression, and cytokine production. This would facilitate reader understanding and better highlight conclusions in each area.

2. The authors should provide a more comprehensive and contextualized discussion, relating their findings to other studies in the field. Exploring how their results align or differ from previous research could help provide a broader context for understanding the role of P. gingivalis in endothelial response.

3. Despite mentioning some limitations, the authors can detail the possible study restrictions further, such as using different cell types, bacterial strains, or experimental techniques. This would further validate their results and provide insights for future research.

4. When the authors mention discrepancies between their findings and previous studies, it is useful to offer hypotheses or possible explanations for these differences. This is important to provide valuable insights into the nuances of interactions between P. gingivalis and endothelial cells.

5. The authors should expand the discussion section, suggesting possible directions for future research. Identifying knowledge gaps or specific aspects that need further investigation would help solidify the importance of their study in advancing the field.

6. The conclusions need to be reformulated. While summarizing the findings, the discussion could benefit from clearer and more direct conclusions about the impact of these results on the overall understanding of the relationship between P. gingivalis and endothelial dysfunction.

Author Response

Dear reviewer.

The following is a response to the reviewers' corrections and suggestions. We thank them for their time and comments; we are convinced that it will improve the article for publication. All corrections are in red in the text of the article.

Responses to Reviewer # 2

Elaborate further on how exactly COX-2, eNOS, and vWF genes influence endothelial function and how their alterations can lead to ED. This could help strengthen the link between genes and atherosclerosis.

Response:  We agree with the reviewer! We have expanded more on this topic in the introduction.

Clearly highlight the specific objective of this study, which is to evaluate the response of HCAECs to P. gingivalis W83 in a 3D cell culture model with collagen compared to the traditional 2D culture model.

Response: The objective of this work was highlighted at the end of the introduction.

Expand the discussion on the relationship between P. gingivalis and endothelial dysfunction. Explain in more detail how the presence of P. gingivalis is directly associated with endothelial dysfunction, highlighting the exact mechanisms by which this bacterium promotes inflammation and atherosclerotic plaque formation.

Response: Thank you for your comment. We have expanded the discussion about this topic.

Explicitly reinforce the limitations of previous cell culture models using HUVECs, emphasizing the importance of using HCAECs in a more representative model of the atherosclerosis context.

Response: we have described some limitations of HUVEC models concerning cellular variable response and strengthened the importance of HCAEC culture. Introduction.

Emphasize the benefits of 3D models. Explain in more detail how the 3D model with collagen allows for a more accurate representation of the microenvironment and cell-matrix interactions, highlighting why it is crucial to study endothelial responses in a more realistic environment.

Response: We have introduced the aspects relevant to  HCAEC 3D support with collagen and accounted for the aspects of cellular and endothelial vascular biology. Introduction.

Methods

The preparation procedures seem detailed and well-defined, allowing for experiment replication. However, detailed characterization of collagen matrices was not provided in this excerpt, despite referencing a previous study for this purpose. It would be advisable to include more details about this characterization in this section.

Response: Thank you for this comment; we have introduced more details about the characterization of collagen matrices.

The authors should provide a more comprehensive and contextualized discussion, relating their findings to other studies in the field. Exploring how their results align or differ from previous research could help provide a broader context for understanding the role of P. gingivalis in endothelial response.

Response: we agree with the reviewer! We have done a more understandable discussion about this topic.

The bacterial culture conditions are well-described, indicating the medium, concentrations, and incubation conditions. However, quantitative data on culture purity are not provided. How did the authors assess this? It would be helpful to include information on contamination, if any, and the percentage of purity.

Response:

Thank you very much for the clarification; this point will be clarified in the methods item.

In the experiments, the purity of the cultures was always checked; all were carried out under sterile conditions, and the controls confirmed the purity of the bacterial inoculum.

The quantification of the P. gingivalis inoculum is described, indicating quantification methods by optical density. However, presenting numerical results of these optical densities would allow for a clearer assessment of bacterial concentration. The evaluation of bacterial viability after 24 hours of stimulation in 2D and 3D cultures is a good indicator, but quantitative results of this viability are not mentioned.

Response:  Thank you very much for your observation; it is clarified in the text that the viability is 100% after 24 hours of carrying out the stimuli.

The methods for culturing human coronary artery endothelial cells (HCAECs) are well-detailed, specifying culture conditions and cell density. However, it is not mentioned whether the experiments were blinded or randomized to minimize bias

Response:  The different items of the methodology corresponding to each technique performed indicate that "Each experimental condition was analyzed in duplicate for three independent experiments," as is indicated for this type of study. It does not apply to blinded or randomized studies.

Results

When mentioning statistically significant or nonsignificant results (p < 0.05 or p > 0.05), it is essential to include the name of the statistical test used to provide a solid basis for conclusions.

Response:  The "Data analysis" methodology item indicates that after performing Shapiro-Wilks normality tests, Kruskal-Wallis tests with Pos Hoc Mann-Whitney U tests were applied to all analyses. Therefore, all significant p correspond to these statistics.

Discussion

The authors should expand the discussion section, suggesting possible directions for future research. Identifying knowledge gaps or specific aspects that need further investigation would help solidify the importance of their study in advancing the field.

Response:  Thank you for this comment. We have addressed future spect to clarify interactions between P. gingivalis and endothelial cells using 3D models that reflect the microenvironment closer to the natural context of endothelial vascular.

Despite mentioning some limitations, the authors can detail the possible study restrictions further, such as using different cell types, bacterial strains, or experimental techniques. This would further validate their results and provide insights for future research

Response:  Some other complementary limitations were added to the discussion.

When the authors mention discrepancies between their findings and previous studies, it is useful to offer hypotheses or possible explanations for these differences. This is important to provide valuable insights into the nuances of interactions between P. gingivalis and endothelial cells.

Response:  Thank you for your suggestion. We have expanded this topic and given one better discussion.

The conclusions need to be reformulated. While summarizing the findings, the discussion could benefit from clearer and more direct conclusions about the impact of these results on the overall understanding of the relationship between P. gingivalis and endothelial dysfunction.

Response: The conclusion was reformulated to account for the main results and their impact.

Round 2

Reviewer 2 Report

Comments and Suggestions for Authors

The authors made significant improvements to the quality of the manuscript, incorporating all the proposed suggestions. The work is now in an acceptable form.